# Brain signatures of a multiscale process of sequence learning in humans

Maxime Maheu[1,2]\*, Stanislas Dehaene[1,3], Florent Meyniel[1]\*

[1]Cognitive Neuroimaging Unit, CEA DRF/JOLIOT, INSERM, Université Paris-Sud, Université Paris-Saclay, NeuroSpin center, Gif-sur-Yvette, France; [2]Université Paris Descartes, Sorbonne Paris Cité, Paris, France; [3]Collège de France, Paris, France

**Abstract** Extracting the temporal structure of sequences of events is crucial for perception, decision-making, and language processing. Here, we investigate the mechanisms by which the brain acquires knowledge of sequences and the possibility that successive brain responses reflect the progressive extraction of sequence statistics at different timescales. We measured brain activity using magnetoencephalography in humans exposed to auditory sequences with various statistical regularities, and we modeled this activity as theoretical surprise levels using several learning models. Successive brain waves related to different types of statistical inferences. Early post-stimulus brain waves denoted a sensitivity to a simple statistic, the frequency of items estimated over a long timescale (habituation). Mid-latency and late brain waves conformed qualitatively and quantitatively to the computational properties of a more complex inference: the learning of recent transition probabilities. Our findings thus support the existence of multiple computational systems for sequence processing involving statistical inferences at multiple scales.
DOI: https://doi.org/10.7554/eLife.41541.001

## Introduction

From seasons' cycle to speech, events in the environment are rarely independent from one another; instead they are often structured in time. For instance, dark clouds often precede rain. Detecting those hidden temporal structures is useful for many cognitive functions (*Lashley, 1951*). For instance, by building correct expectations of the future, one can then process sensory information more quickly and/or more efficiently (*Summerfield and de Lange, 2014*; *Melloni et al., 2011*; *Ekman et al., 2017*) or adapt behavior by selecting the best course of actions (*Skinner, 1953*). In humans, the extraction of structures over time is also crucial in language acquisition (*Saffran et al., 1996*; *Endress and Mehler, 2009*).

Sequences of observations can be described at different levels of abstraction (*Dehaene et al., 2015*). The most basic level of coding used to represent sequential inputs is the encoding of statistical regularities, such as *item frequency* and *transition probabilities*. Many experiments demonstrate that the brain possesses powerful statistical learning mechanisms that extract such regularities from sequential inputs (*Armstrong et al., 2017*; *Santolin and Saffran, 2018*). Importantly, the learnt statistics can serve as building blocks for more elaborate codes. For instance, in infants, learning the *transition probabilities* between syllables seems to be a building block on top of which words and syntactic tree structures are built (*Saffran et al., 1996*; *Chomsky and Ronat, 1998*). However, the exact manner through which the brain extracts such statistics remains unknown. Here, we ask two basic questions: What statistics are estimated by the brain? Over which timescale are such computations performed, that is which observations, from recent to remote ones, are incorporated in the learning process? As we shall see, there is no single answer to those questions because different brain processes estimate different statistics computed over different timescales. However, we show how learning models, fitted to brain responses, can help tease apart those processes.

**\*For correspondence:**
maheu.mp@gmail.com (MM);
florent.meyniel@gmail.com (FM)

**Competing interests:** The authors declare that no competing interests exist.

Several statistics can indeed be used to describe a given sequence. To illustrate this point, consider the following sequence: AAAAABBBBBBAA. A first level of description is the frequency of each item (A and B). In this example: $p(A) = 1 – p(B) = 7/13$. The brain is sensitive to the *frequency of items* in both visual (*Grill-Spector et al., 2006*) and auditory sequences (*Näätänen et al., 2007*; *Garrido et al., 2009*). But note that *item frequency* (IF) ignores the order in which observations are received. By contrast, *alternation frequency* (AF) inspects whether successive observations are identical or different, a statistic to which the brain is also sensitive in visual (*Summerfield et al., 2008*; *Teinonen et al., 2009*; *Bulf et al., 2011*; *Summerfield et al., 2011*) and auditory sequences (*Todorovic et al., 2011*; *Todorovic and de Lange, 2012*). In our example, $p(\text{alternation}) = 1 – p(\text{repetition}) = 2/12$. Note that *alternation frequency* ignores the nature of repetitions and alternations, as it treats similarly the repetitions A → A and B → B and similarly the alternations A → B and B → A. Such dependencies can only be captured by *transition probabilities* (TP), which specify what stimulus is likely to be observed (an A or a B) given the context. In our example, $p(A|B) = 1 – p(B|B) = 1/6$ and $p(B|A) = 1 – p(A|A) = 1/6$. Importantly, TP entail information regarding the *frequency of items* (IF), their co-occurence (AF), and their serial order (e.g. $p(A|B) \neq p(B|A)$). This makes TP the simplest statistical information that genuinely reflects a sequential regularity. There is evidence that TP are monitored by human adults (*Domenech and Dreher, 2010*; *Meyniel et al., 2015*; *Mittag et al., 2016*; *Meyniel and Dehaene, 2017*; *Higashi et al., 2017*) and infants (*Marcovitch and Lewkowicz, 2009*; *Lipkind et al., 2013*), macaque monkeys (*Meyer and Olson, 2011*; *Ramachandran et al., 2016*), rodents (*Yaron et al., 2012*) and even songbirds (*Takahasi et al., 2010*; *Markowitz et al., 2013*; *Lipkind et al., 2013*; *Chen and Ten Cate, 2015*). Those three different statistics (IF, AF, TP) are, however, often studied separately. This may cause confusion, because some (IF and AF) are embedded in others (TP). In addition, the correct identification of those statistics depends on the timescale over which they are computed, which may vary between studies. For instance, if the following elements are presented sequentially: AAAAABBBBBBAA..., the underlined B will appear surprising with respect to both IF (a B after many As) and AF (an alternation after many repetitions), whereas the underlined A will not appear surprising to AF (it follows the tendency to repeat), and comparatively more surprising to IF since at the scale of a few observations this A is quite rare relative to the many preceding Bs.

Our second question is therefore about timescales: which history of observations is used to inform the estimated statistics? Previous studies showed that the brain is particularly sensitive to the recent history of observations both in visual (*Huettel et al., 2002*) and auditory sequences (*Squires et al., 1976*; *Kolossa et al., 2012*). Moreover, brain responses elicited by each observation indicate the existence of different statistical expectations arising in parallel from distinct observations' histories (*Ulanovsky et al., 2004*; *Kiebel et al., 2008*; *Bernacchia et al., 2011*; *Ossmy et al., 2013*; *Meder et al., 2017*; *Scott et al., 2017*; *Runyan et al., 2017*). There exists a continuum of timescales, but for clarity and following previous works (*Squires et al., 1976*; *Kolossa et al., 2012*; *Meyniel et al., 2016*), we will refer to statistics being estimated using the most recent observations only, as *local statistics*, and we term *global statistics* the ones whose estimation takes into account all observations.

Taken altogether, the prior results quoted so far suggest that the brain is sensitive to different statistics (IF, AF, TP) and different timescales of integration (from local to global). Yet those conclusions remain unclear because those effects were investigated using different techniques (e.g. reaction times, EEG, fMRI), in different modalities (using sounds, faces, geometrical shapes, ...) and with only a subset of all possible conditions (e.g. by manipulating IF but not AF). We recently proposed that those seemingly disparate findings might in fact all result from an estimation of TP on a local timescale (*Meyniel et al., 2016*). In this view, (i) sensitivity to global, block-level, statistics results from an average of local expectations, and (ii) sensitivity to simpler statistics (IF and AF) is explained by the fact that they are embedded in the space of TP. Here, we will explore whether different brain signatures reflect computations of different statistics and different timescales, and the extent to which they correspond to our proposed local TP model.

# Results

## Experimental design

In order to address those questions, we designed a new MEG experiment in which binary auditory sequences comprising two sounds, denoted A and B, were presented to human subjects (see *Figure 1A*). Different statistics were used to generate these stochastic sequences (see *Figure 1B*). In the frequency-biased condition, one item (B) was overall more frequent than the other. In the alternation-biased condition, alternation (i.e. pairs AB and BA) was overall more frequent than repetition (i.e. AA and BB). In the repetition-biased condition, the reverse was true. Finally, in the fully stochastic condition, all the events were equally likely to happen (i.e. as many As than Bs and as many AA, than AB, BA, BB).

## A quantitative account of learning

It is important to distinguish the statistics that are used to generate the sequences and the statistics that are inferred by the brain. In order to characterize the inference process, we designed three families of learning models, each corresponding to the inference of a different statistics: IF, AF and TP. Each of the sequences presented to subjects corresponds to a different bias in generative TP, and it can be described from the view point of each model (IF, AF, TP). *Figure 1D* shows an example for each of the 12 possibilities (three models times four sequences generated with a different bias).

Within each family of model (one per statistics that is estimated), we further distinguished between models on the basis of their timescale of integration. An extreme case is global integration:

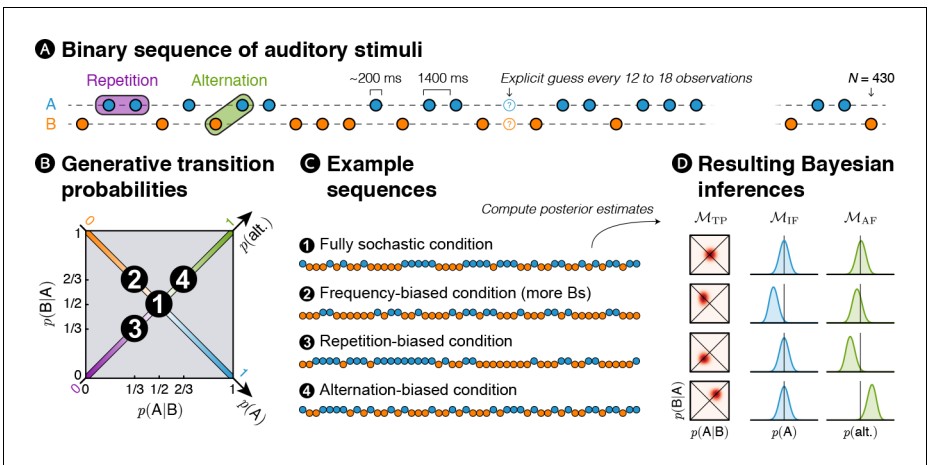

**Figure 1.** Experimental design: *transition probabilities* induce orthogonal variations of *item frequency* and *alternation frequency*. (**A**) Human subjects were presented with binary sequences of auditory stimuli. Each sequence was composed of a unique set of syllables (e.g. A = /ka/ and B = /pi/) presented at a relatively slow rhythm (every 1.4 s) and occasionally interrupted by questions asking subjects to predict the next stimulus. (**B**) Stimuli were drawn from fixed generative *transition probabilities* p(A|B) and p(B|A) that varied across four blocks. Those *transition probabilities*, in turn, determined *item frequency* p(A) and *alternation frequency* p(alt.). (**C**) Example sequences derived from generative *transition probabilities* used in each condition. The resulting sequences were either fully stochastic, biased toward one of the stimuli (here Bs), or biased toward repetitions or alternations. (**D**) Learning models were applied to these sequences: a model learning *transition probabilities* (TP), one learning the *frequency of items* (IF) and one learning the *frequency of alternations* (AF). Note that only the TP model can discriminate all four conditions; the IF model is blind to biases toward repetitions or alternations, while the AF model is blind to biases in the balance between stimuli.

DOI: https://doi.org/10.7554/eLife.41541.002

The following figure supplements are available for figure 1:

**Figure supplement 1.** Diagnostic value of the experimental design.

DOI: https://doi.org/10.7554/eLife.41541.003

**Figure supplement 2.** Brain responses evoked by the auditory stimuli.

DOI: https://doi.org/10.7554/eLife.41541.004

all observations are used equally in the inference process (see *Figure 1D*). A more realistic possibility is local integration by means of leaky accumulation (*Gerstner and Kistler, 2002*): previous observations are progressively discounted, giving a larger weight to the most recent observations in the inference process.

All models estimate the current statistics based on a (local or global) count of observations by means of standard Bayesian inference (see Materials and methods). Using Bayesian inference (again), the estimated statistics are turned into a prediction about the next stimulus. In that framework, surprise results from the discrepancy between the prediction and the actual observation that is received. More formally, surprise is quantified as the negative logarithm of the (estimated) probability of the actual observation (*Shannon, 1948*): observations that are deemed less probable produce larger surprise.

Many stimulus-evoked MEG responses are typically thought of as surprise signals, or equivalently, prediction error signals. According to the predictive coding hypothesis (*Rao and Ballard, 1999*; *Friston, 2005*; *Spratling, 2017*), prediction error is an important quantity used to revise our beliefs (*Friston and Kiebel, 2009*) or simply to perceive the world (*Rao and Ballard, 1999*; *Lee and Mumford, 2003*). Such surprise signals dominate evoked responses measured with MEG and EEG, in particular in the auditory domain (*Heilbron and Chait, 2018*), and are thus fairly easy to quantify. We therefore adopted a reverse-engineering approach, comparing MEG signals to theoretical surprise levels computed from different learning models. Based on this comparison, we aimed to determine the most likely inference performed by the brain among those that we have hypothesized. We compared different learning models that differ both in the statistics (IF, AF and TP) and the timescale of integration (parameterized by $\omega$, we tested 55 values in total). We adopted a design that allows the identification of both the statistics that are inferred by the brain, and the timescale of integration (see *Figure 1—figure supplement 1*).

## Description of auditory-evoked responses

We first verified that our paradigm elicited typical auditory responses. The grand-averaged MEG responses evoked by the sounds revealed four distinct canonical components (see *Figure 1—figure supplement 2*): three early responses (M50, M150 and M250 components) characterized by topographies plausibly arising from bilateral auditory cortex (*Chait et al., 2007*; *Strauss et al., 2015*) and a late response associated with the classical P300/slow-wave topography (*Wacongne et al., 2011*).

In order to investigate whether those brain responses were modulated by the statistical regularities of the auditory sequences, we explored two timescales at which those regularities may emerge in our experiment: global and local. Our general approach proceeds as follows: (1) searching effects of global statistical regularities, (2) investigating the interplay between local and global statistical regularities in order to follow up and expand previous historical studies (in particular: *Squires et al., 1976*) that focused on late evoked responses and (3) performing a systematic analysis, independent of any prior assumption regarding the timing (what post-stimulus latency), spatial location (which sensors) and properties (which effect, local or global, and of which statistics) of the effects.

## Sensitivity to global, block-level, statistics

We first examined how these auditory-evoked responses were modulated by global, block-level statistics within each of the four conditions. By design, three out of these four experimental conditions induce a bias in the global statistics, such that some events are overall more likely to occur than others. To investigate the brain sensitivity to such global biases, we contrasted MEG responses evoked by rare events with frequent ones. We report results corrected for multiple comparison across time points and sensors (the spatio-temporal clusters that survive correction at $p < 0.05$ at the test level and $p < 0.05$ at the cluster level, see Materials and methods and *Figure 2—Source data 1*).

In the frequency-biased condition, rare A items were overall less frequent than frequent B items and, as expected, rare items elicited larger responses than frequent ones (significant between 227 and 910 ms post-sound onset, see *Figure 2A* and *Figure 2—figure supplement 1A*). In the repetition-biased condition, repetitions, that is XX = AA or BB, were overall more frequent than alternations, that is XY = AB or BA; brain signals also reflected this bias, showing increased activity for rare alternations as compared to frequent repetitions (significant between 391 and 633 ms post-sound

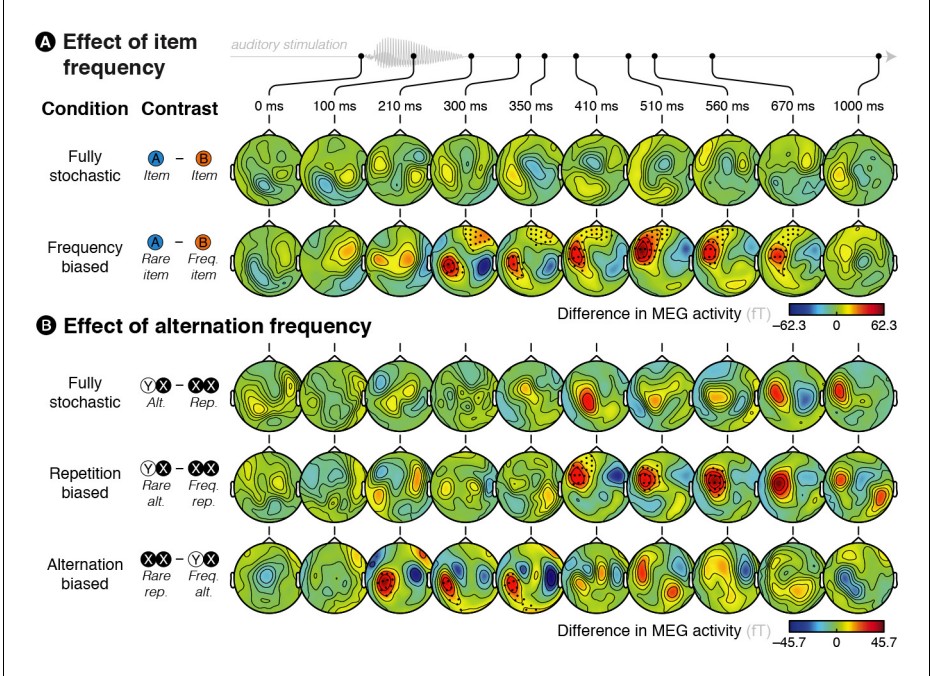

**Figure 2.** Brain responses are modulated by global statistics. (**A**) Difference in brain activity evoked by the two different items in the fully stochastic (in which both stimuli are equiprobable) and frequency-biased (in which one of the two stimuli is more frequent than the other) conditions. (**B**) Difference in brain activity evoked by alternations and repetitions in the fully stochastic (in which repetitions and alternations are equiprobable), repetition-biased (in which repetitions are more frequent than alternations) and alternation-biased (in which alternations are more frequent than repetitions) conditions. X/Y denotes a pooling of both stimuli together, XX (or YY) thus denotes repetitions of A or B (i.e. AA and BB pooled) while XY (or YX) denotes alternations between A and B (i.e. AB and BA pooled). We report the difference in the MEG responses elicited by a given item (X = A or B) when it is preceded by the same (XX = AA or BB) versus a different (YX = BA or AB) item. Sensors marked with a black dot showed a significant difference ($p < 0.05$ at the test level and $p < 0.05$ at the cluster level).
DOI: https://doi.org/10.7554/eLife.41541.005

The following source data and figure supplements are available for figure 2:

**Source data 1.** Violation of global statistics.
DOI: https://doi.org/10.7554/eLife.41541.008
**Figure supplement 1.** Time courses of the differences between frequent and rare events in the biased conditions.
DOI: https://doi.org/10.7554/eLife.41541.006
**Figure supplement 2.** Contrasts between surprise responses across conditions.
DOI: https://doi.org/10.7554/eLife.41541.007

onset, see *Figure 2B* and *Figure 2—figure supplement 1B*). In the alternation-biased condition, in which alternations are overall more frequent than repetitions, we found a reversal of the effect compared to the repetition-biased condition. Brain signals were larger for rare repetitions than frequent alternations (significant between 176 and 371 ms post-sound onset, see *Figure 2B* and *Figure 2—figure supplement 1B*). Finally, in the fully stochastic condition, there was no global bias regarding *item* or *alternation frequencies*: there were as many As as Bs and as many repetitions as alternations. Accordingly, no significant difference was found, neither for the A – B contrast (see first row in *Figure 2A*) nor for the repetition – alternation one (see first row in *Figure 2B*). Between-conditions comparisons show that differences between items or between alternation and repetition are indeed stronger when there exists a global statistical bias than in the fully stochastic condition (see *Figure 2—figure supplement 2*).

## Sensitivity to local, history-dependent, statistics

Beyond and above those block effects, due to their stochastic nature, the sequences also contain biases in the local statistics (e.g. a sudden surge of A sounds). Previous findings have shown that some brain signals, notably late ones, reflect a sensitivity to such local regularities on top, or even in the absence of global regularities (*Squires et al., 1976*; *Kolossa et al., 2012*). In a seminal publication, *Squires et al. (1976)* introduced an analysis that allows a clear visualization of those effects. They constructed a spatial filter to project sensor data in a specific (late) time window onto one dimension that maximized the difference between globally rare and frequent items. They then used this same filter to plot the brain response to all possible local patterns of stimuli, thereby revealing prominent effects of local statistics, such as differences between locally rare/frequent items, and between locally rare/frequent alternations and repetitions. Here, we extended their analysis: we also inspected local effects in the repetition- and alternation-biased conditions, while previous studies (*Squires et al., 1976*; *Kolossa et al., 2012*) only had frequency-biased and fully stochastic conditions. We used the group-averaged topographical difference between continuing and violating streaks as spatial filters in a late time window from 500 to 730 ms (see *Figure 3—figure supplement 1* for the exact topographies used) but unlike previous studies, we defined and applied these filters in a cross-validated manner to avoid circular analysis (*Kriegeskorte et al., 2009*). Following *Squires et al. (1976)*, we then plotted the spatially filtered response to all possible local patterns up to length 3, in the same late time window (from 500 to 730 ms) for all conditions, in the form of a

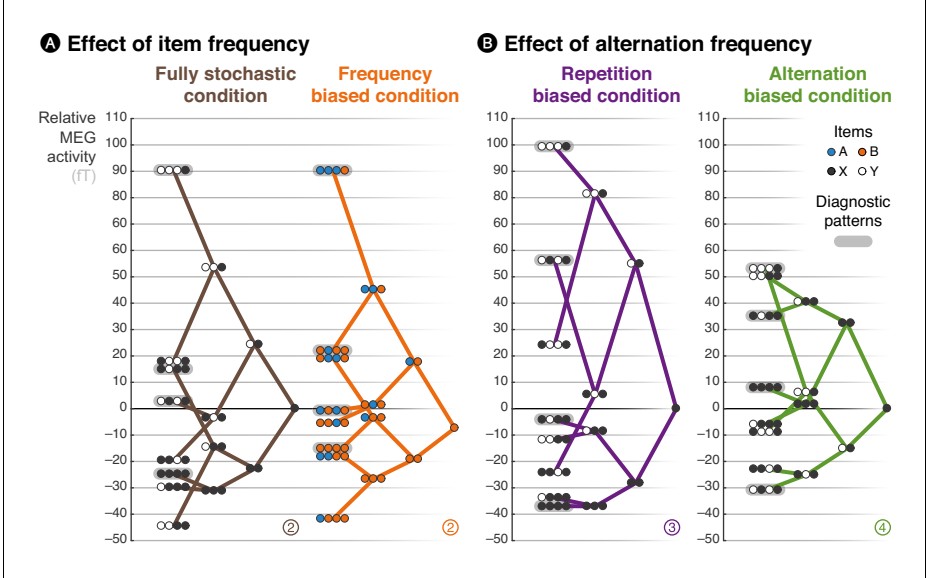

**Figure 3.** Brain responses are modulated by local statistics. The brain response to a given observation is plotted as a function of the recent history of events. We connected the possible extensions from shorter to longer patterns, such that the data are represented as a 'tree'. Each of these trees corresponds to an experimental condition. At each node of a tree, the circles should be read from left to right and denote the corresponding patterns; for instance, the pattern AAAB shows the activity level elicited by the item B when it was preceded by three As. X and Y denote the pooling of both stimuli in conditions in which both stimuli are equiprobable. In the frequency-biased condition, we report the activity evoked by item B, the most frequent stimulus. Activity levels across sensors were averaged using a topographical filter within a late time window (from 500 to 730 ms) post-stimulus onset. The topographical filters were obtained by contrasting rare and frequent patterns (e.g. XYXX – XYXY in the alternation-biased condition). The filters are shown in *Figure 3—figure supplement 1*; the small, circled and colored numbers at the bottom of each tree serve as identifiers. We defined and applied the filters using a cross-validation approach to ensure statistical independence (see Materials and methods).
DOI: https://doi.org/10.7554/eLife.41541.009

The following figure supplement is available for figure 3:

**Figure supplement 1.** Responses to the violation of local patterns used as spatial filters.
DOI: https://doi.org/10.7554/eLife.41541.010

tree whose structure reflects the effect of local statistics on the amplitude of MEG signals (see *Figure 3*).

Note that this late time window (from 500 to 730 ms) was selected in order to follow up and expand the seminal study of *Squires et al. (1976)* which focused on late (EEG) brain responses. Here, the exact time window was not optimized to show the clear tree structure seen in *Figure 3*, or the strong similarity between observed data and the model reported in *Figure 4*, but selected a priori (see *Figure 3—figure supplement 1*).

This graphical representation summarizes several interesting aspects of the data, of which we provide a statistical analysis below together with the predictions of our model. First, alternating pairs (i.e. XY) elicited larger signals than repeating ones (i.e. XX) in all conditions except the alternation-biased condition, in which this order was reversed. Second, the violation of a repeating streak (i.e. XXXY) always produced large activity, no matter of the experimental condition. This is particularly interesting in the alternation-biased condition, since such violations are actually expected in this case. Last, the continuation of an alternating pattern (i.e. XYXY) produced intermediate activity in both the fully stochastic and frequency-biased conditions, but low activity in the alternation-biased

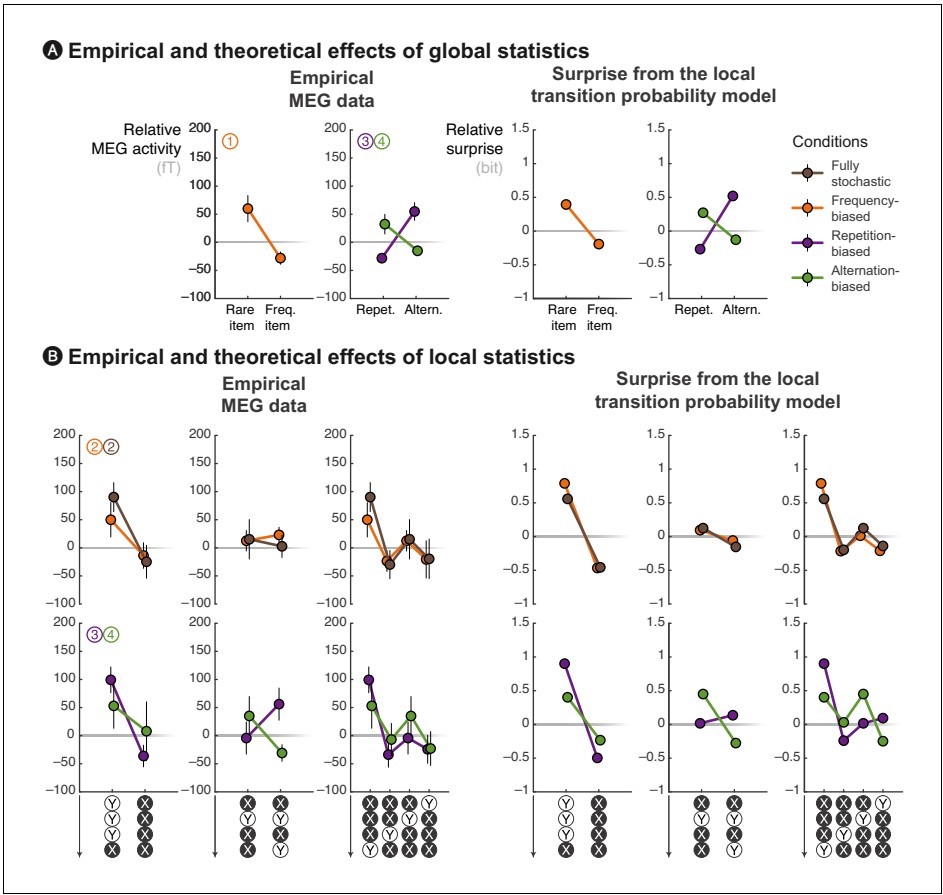

**Figure 4.** The local *transition probability* model accounts for modulations of late MEG signals. Observed MEG signals and theoretical surprise levels predicted by the *transition probability* model (with a local integration, $\omega = 6$) in response to (**A**) globally rare events and to (**B**) violation or continuation of local patterns. MEG signals correspond to activity levels across sensors were averaged using a topographical filter (see *Figure 3—figure supplement 1*) within a late time window (from 500 to 730 ms) post-stimulus onset. The patterns reported here correspond to the diagnostic ones that are highlighted in *Figure 3*. See *Figure 4—figure supplement 1* for theoretical predictions by other models.

DOI: https://doi.org/10.7554/eLife.41541.011

The following figure supplement is available for figure 4:

**Figure supplement 1.** Qualitative account of experimental effects by rival models.

DOI: https://doi.org/10.7554/eLife.41541.012

condition and high activity in the repetition-biased condition. Altogether these results confirm and extend what has been observed in the past: the effect of local statistic onto brain responses are modulated by the global *frequency of items* in the block, but also by the global *frequency of alternations*.

## Comparison of MEG signals with proposed learning models

We now turn to a quantitative analysis of late brain responses (from 500 to 730 ms, as in *Figure 3*) with cross-validated topographical filter (see *Figure 3—figure supplement 1*) and we compare the most diagnostic effects with the predictions of different learning models. We show that a parsimonious model, the estimation of local TP ($p(A|B) = 1 - p(B|B)$ and $p(B|A) = 1 - p(A|A)$, TP model) predicts the effects of local and global statistics observed in brain responses (see *Figure 4*) and that those effects are incompatible with two other models (see *Figure 4—figure supplement 1*), learning either the local *frequency of items* (i.e. $p(A) = 1 - p(B)$, IF model) or the local *frequency of alternations* between these items (i.e. $p(alt.) = 1 - p(rep.)$, AF model).

We begin with the effects of global, block-level statistics. This analysis supplements findings reported in *Figure 2*, but now with cross-validated topographical filters in order to facilitate comparison with the models. *Figure 4A* shows the late MEG responses evoked by rare and frequent events in the three biased conditions. First, in the frequency-biased condition, rare items evoke significantly stronger MEG signals than frequent ones ($t_{17} = 2.57$, $p = 0.020$), similarly to the surprise of the TP model. Note in *Figure 4—figure supplement 1A* that the alternative IF model also captures this difference and that the AF model shows by comparison a weaker effect. Indeed, in the frequency-biased condition, there exists a weak bias in favor of repetitions, $p(rep.) = 5/9$, which is smaller than the frequency bias, $p(B) = 2/3$.

Second, in the repetition-biased condition, rare alternations evoked stronger signals than frequent repetitions ($t_{17} = 2.97$, $p = 8.63 \cdot 10^{-3}$). Conversely, in the alternation-biased condition, rare repetitions elicit stronger signals than frequent alternations; however, this difference is expectedly smaller than in the repetition-biased condition and it was indeed not significant ($t_{17} = 1.33$, $p = 0.20$). This reversal of repetition/alternation effect in alternation-/repetition-biased conditions is significant when tested as an interaction in an ANOVA ($F_{1,17} = 17.26$, $p = 6.64 \cdot 10^{-4}$). Such an interaction is fully predicted by the TP model (see *Figure 4A*). Note that the AF model, but not the IF model, shows such interaction (see *Figure 4—figure supplement 1A*). To summarize, the TP model is the only model that predicts the observed effects of global statistics onto MEG responses across all conditions.

We now turn to the effects of local statistics in short series of stimuli. *Figure 4B* shows several effects of the violation or continuation of short patterns of stimuli on MEG responses in the different conditions. First, we compared MEG responses elicited by the continuation and violation of a repeating streak. The violation of repeating streaks evoked significantly larger signals than their continuation in the fully-stochastic ($t_{17} = 3.12$, $p = 6.21 \cdot 10^{-3}$) as well as in the frequency-biased condition ($t_{17} = 2.34$, $p = 0.032$). As expected, they also induced significantly stronger brain signals in the repetition-biased condition ($t_{17} = 3.49$, $p = 2.79 \cdot 10^{-3}$). There was a similar trend in the alternation-biased condition, but non-significant ($t_{17} = 0.63$, $p = 0.54$) probably because of the conflict between global and local statistics. The ANOVA did not reveal a significant interaction ($F_{1,17} = 1.60$, $p = 0.22$) between repetition-/alternation-biased conditions and the continuation/violation of a repeating streak. These local effects are accounted for by all models (see *Figure 4B* and *Figure 4—figure supplement 1B*).

Second, we compared MEG responses elicited by the continuation and violation of an alternating streak. The violation and continuation evoked similar brain signals in both the fully-stochastic ($t_{17} = 0.27$, $p = 0.79$) and the frequency-biased ($t_{17} = 0.47$, $p = 0.65$) conditions. The difference was not significant in the repetition-biased ($t_{17} = -1.39$, $p = 0.18$) and alternation-biased ($t_{17} = 1.62$, $p = 0.13$) conditions, but interestingly, it appeared with opposite signs. The ANOVA indeed revealed a trend regarding the interaction between repetition-/alternation-biased conditions and the continuation/violation of an alternating streak ($F_{1,17} = 3.69$, $p = 0.07$). This local effect of alternations is predicted by the TP and AF models, but not by the IF model (see *Figure 4B* and *Figure 4—figure supplement 1B*).

Third, we compared MEG responses elicited by the same observations received in different orders. Since a local integration assigns gradually smaller weights to more remote observations, it

indeed predicts an effect of the order in which observations are received. We compared four local patterns all composed of 3 Xs and 1 Y but differing in the position of the Y (i.e. first, second, third or last position). The profile of difference in MEG activity evoked by these different patterns was qualitatively similar to what was predicted by the TP model, but not the IF nor the AF ones (see *Figure 4B* and *Figure 4—figure supplement 1B*).

## Computational characterization of MEG signals

The previous analyses, like previous studies, focused on the effect of very short patterns (here, up to 3 or 4 observations) and specific brain responses (here, a late component). In principle, however, we can analyze the effect of any timescale of integration onto brain responses of any given latency. To extend the results obtained so far, we therefore run a more systematic analysis aiming at characterizing quantitatively, from an information-theoretic viewpoint, the trial-by-trial fluctuations in MEG responses recorded at any latency post-stimulus onset (up to 1 s) and any sensor.

To achieve this goal, we performed trial-by-trial regressions for each subject between MEG responses (separately for each sensor and each time point between –250 ms to 1 s post-sound) recorded in all conditions and the theoretical surprise levels of various learning models described above. An example regression is shown in *Figure 5A*. Note that a similar model-based approach to brain electrophysiological signals has already been adopted in the past (*Strange et al., 2005*; *Lieder et al., 2013a*; *Lieder et al., 2013b*; *Mars et al., 2008*).

The maximum proportion of variance over models and parameter was averaged across sensors (see *Figure 5B* and *Figure 5—video 1*) and it reveals three distinct time windows of interest: an early one (60 to 130 ms, peak at 70 ms), an intermediate one (160 to 320 ms, peak at 250 ms) and a late one (460 to 625 ms, peak at 500 ms). The $R^2$ topography in each time window showed peaks over fronto-temporal sensors (see *Figure 5C*) as expected for auditory stimuli. This topography was highly bilateral for the first window, slightly left lateralized for the second one and strongly left lateralized for the last one. Note that these topographies are similar to the ones obtained in previous model-free analyses, indicating that the model-based analysis identified similar effects but without any a priori about their spatial or temporal properties.

We then performed a Bayesian model selection (BMS, see Materials and methods; *Stephan et al., 2009*) to identify the learning model that best corresponds to each of the identified time windows. In the first time window, the model learning IF was deemed the best with high posterior probability ($p(M_{IF}|y) = 0.978$) and high confidence ($\varphi = 1$); thus, this early response was best characterized as an adaptation to individual frequent stimuli. By contrast, in the second ($p(M_{TP}|y) = 0.781$, $\varphi = 0.994$) and the third ($p(M_{TP}|y) = 0.993$, $\varphi = 1$) time windows, the observer learning TP best explained the MEG signals (see *Figure 5D*). We also performed a hierarchical Bayesian model selection (*Rigoux et al., 2014*) over the four different conditions that found high confidence ($\varphi = 1$) that the same models are likely to underlie all conditions, indicating that the results are not driven by one condition only.

Finally, in order to draw inference about the timescale of integration of each of these three signatures, we performed a Bayesian model averaging (BMA, see Materials and methods). This analyses optimally combines the different possible models, weighted by their uncertainty in order to obtain subject-specific posterior distributions over parameter values. The result shows that the later the MEG time windows, the shorter the timescale of integration (linear fit using the mean of subject-specific posterior distributions of $\omega$: $t_{17} = -3.00$, $p = 0.008$). At the group level (see *Figure 5E*), the first window was best characterized by a global integration ($\omega = \infty$), the second and third by much more local integration ($\omega = 13$ and 6, respectively, maximum a posteriori values; that is a half-life of 9.01 and 4.16 observations, respectively).

For completeness, *Figure 5—figure supplement 1* shows the posterior probability of each model and timescale of integration in continuous time rather than in specific time windows.

## Discussion

In this study, we found a series of brain signals that covary with the online inference of the statistical properties of auditory sequences of stimuli. Late brain signals qualitatively and quantitatively conformed to the inference of *transition probabilities*, while earlier brain waves denoted a sensitivity to simpler statistics, such as the *item frequency*. Furthermore, those different signals were

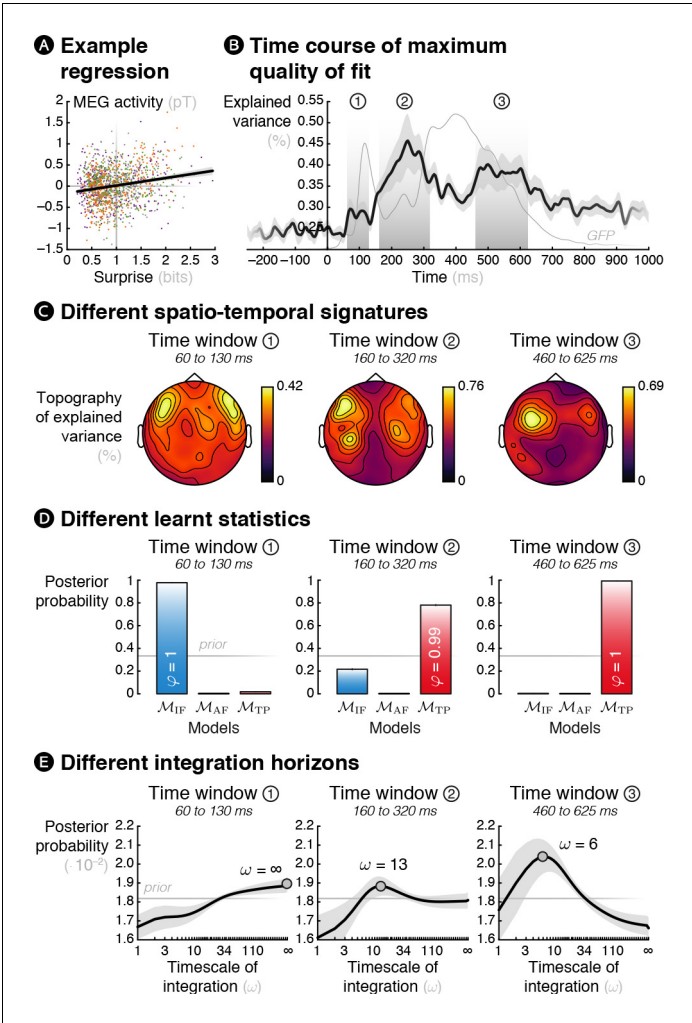

**Figure 5.** Theoretical surprise levels from learning models fitted to MEG signals reveal a multiscale inference process. Mass-univariate (across sensors and time points) trial-by-trial regressions of MEG signals against theoretical surprise levels from learning models learning different statistics with different timescale of integration ($\omega$). (**A**) Example trial-by-trial regression (the colors represent the different conditions), for one subject, at one particular time point, in one particular sensor in the case of *transition probability* learning with $\omega = 16$. (**B**) The maximum $R^2$ over models and parameters was averaged across sensors, and then across subjects. The resulting time course reveals three distinct time windows in which regressions yield a high proportion of explained variance. For reference, the thin grey line represents the global field power evoked by all the sounds (see **Figure 1—figure supplement 2A**). (**C**) Average topography of maximum $R^2$ values in each time window. (**D**) Bayesian model comparison reveals that different models best explain MEG signals in these three time windows: *item frequency* (IF) in an early time window, and *transition probabilities* (TP) in later time windows. (**E**) Bayesian model averaging further reveals that different timescales of integration are involved: slow, global integration for the early time window, and increasingly local integration for the later time windows. These posterior distributions give the probability of $\omega$ given the MEG data. The error shading shows the inter-subject s.e.m.

DOI: https://doi.org/10.7554/eLife.41541.013

The following video and figure supplements are available for figure 5:

**Figure supplement 1.** Time-resolved model comparison.
DOI: https://doi.org/10.7554/eLife.41541.014

**Figure supplement 2.** Characteristics of the free parameter controlling the timescale of integration.
DOI: https://doi.org/10.7554/eLife.41541.015

**Figure 5—video 1.** Movie of $R^2$ topographies.
DOI: https://doi.org/10.7554/eLife.41541.016

characterized by different timescales of integration: late signals were best explained by a very local timescale that progressively discards past observations, while early signals denoted a much longer timescale of integration.

Below, we discuss successively both aspects: the types of statistics that might be inferred by the brain when it is presented with a sequence of observations and the timescales of integration involved in those inferences. But first, we highlight an important distinction between sequence learning experiments. Some studies use sequences structured into *chunks of stimuli*: the sequences are interrupted by short pauses that define chunks of stimuli, thereby cueing subjects about the type of regularities to look for and their timescale. Those chunks are often pairs (*Todorovic et al., 2011*; *Todorovic et al., 2011*; *Meyer and Olson, 2011*; *Meyer et al., 2014*) or longer patterns, for example of 5 stimuli (*Bekinschtein et al., 2009*; *Wacongne et al., 2011*; *Strauss et al., 2015*). In contrast, we used continuous sequences, devoid of chunks of stimuli, very much like in the oddball literature (*Squires et al., 1976*; *Näätänen et al., 2007*). The presence of chunks is crucial for the processing of sequences in the brain (*Dehaene et al., 2015*), as exemplified by the strikingly different results obtained by *Sussman et al. (1998)*, *Sussman and Gumenyuk (2005)* and by *Bekinschtein et al. (2009)* using similar sequences that were, respectively, either continuous or structured into chunks with pauses. We will therefore discuss experiments with chunks and continuous sequences separately.

## Sensitivity to item frequency

Many previous studies have manipulated the *frequency of items* within continuous sequences presented to subjects and reported larger brain responses for rare items compared to frequent ones. In oddball paradigms for instance, globally rare (deviants) items systematically evoke stronger brain signals than frequent (standard) ones (*Näätänen et al., 2007*; *Garrido et al., 2009*; *Garrido et al., 2013*; *Hsu et al., 2015*; *Lecaignard et al., 2015*). On top of the effect of the global *frequency of items*, the seminal work of *Squires et al. (1976)* as well as more recent findings (*Ulanovsky et al., 2004*; *Kolossa et al., 2012*) showed that local regularities also modulate brain signals. In particular *Squires et al. (1976)* showed that the late P300 response evoked by a particular item was modulated by the preceding history of stimuli: the violations of repeating (XXX$\underline{Y}$) or alternating (XYX$\underline{X}$) streaks evoked stronger brain signals than their continuation (XXX$\underline{X}$ and XYX$\underline{Y}$). The modulatory effects of local continuation/violation of local patterns were observed even in a fully random sequence, in which there is no global bias in the *frequency of items*.

Here, we replicated and extended these findings. First, the MEG responses that we recorded were strongly modulated by the global *frequency of items*. Second, despite the limited number of observations in some cases (e.g. $p(AAAB) = (1/3)^3 \cdot (2/3) = 2.5\%$ of all patterns of length four in the frequency-biased condition), we found effects of local patterns both in the fully stochastic (where $p(A) = p(B) = 1/2$) and frequency-biased (where $p(A) = 1 - p(B) = 1/3$) conditions. Furthermore, we unify the effects of both local and global statistics into the single framework of Bayesian inference (*Meyniel et al., 2016*). In that framework the brain is seen as an inference device that seeks to predict future observations, and recorded brain signals are related to theoretical surprise levels elicited by the comparison between predictions and actual observations (*Friston, 2005*). Here, we provide new data in support of our hypothesis that several brain signals, in particular those characterized by a late latency, can be best accounted for as surprise signals corresponding to the learning of *transition probabilities*.

## Sensitivity to alternation frequency

Our hypothesis about an inference of *transition probabilities* makes an additional important prediction, rarely tested in several previous studies of mismatch responses in continuous sequences. Indeed, not only *item frequency*, but also the *frequency of alternations* is embedded in the space of *transition probabilities*. It follows that rare alternations should be more surprising than frequent repetitions and, conversely, that rare repetitions should be more surprising than frequent alternations. Recent studies manipulated the *frequency of alternations* while keeping the *item frequency* fixed and reported that rare alternations indeed induce stronger brain signals than frequent repetitions (using continuous sequences: *Summerfield et al., 2011*; *de Gardelle et al., 2013*; using pairs: *Summerfield et al., 2008*; *Grotheer and Kovács, 2014*, *Grotheer and Kovács, 2015*; *Tang et al.,*

*2018*; *Feuerriegel et al., 2018*). However, in continuous sequence paradigms, the reverse is rarely observed: rare repetitions often fail to induce stronger brain responses than frequent alternations (but see in pair learning paradigms: *Todorovic and de Lange, 2012*; and in pattern learning paradigms: *Strauss et al., 2015*). In most cases, there is only a reduced difference (slightly more signal for frequent alternations than for rare repetitions) rather than a genuine reversal (more signal for rare repetitions than for frequent alternations).

On the contrary, one important finding of our study, in line with predictive coding (*Spratling, 2017*; *Heilbron and Chait, 2018*) and the learning of *transition probabilities*, is that we observe a genuine reversal of brain activity evoked by repetitions and alternations depending on which one is more frequent (by comparing the repetition-biased and alternation-biased conditions). We also extended those results by demonstrating effects of local statistics on brain signals even when there is a bias in the *frequency of alternations*. We report several findings that are predicted by the *transition probability* model. For instance, compared to other patterns, the repeating streak XXXX elicits more signal in the alternation-biased condition than in the repetition-biased condition. Conversely, the alternating pattern XYXY elicits more signal in the repetition-biased condition than in the alternation-biased condition. On the contrary, the violated repeating pattern XXXY always elicits stronger signal than other patterns, no matter the condition. Those subtle effects in the ordering of patterns result from the sometimes conflicting effects of local and global regularities, and critically, they are all predicted by a model according to which the brain attempts to infer *transition probabilities*.

## A general account in terms of transition probability

Previous studies that manipulated either the *frequency of items* or the *frequency of alternations* provide substantial evidence that the brain is sensitive to both types of statistics. However, they left unclear whether their results could be attributed, and possibly better explained, by a sensitivity to a higher-order statistics like the *transition probabilities* that subsume *item* and *alternation frequencies*. Studies that manipulate those two statistics are actually rare. In some of them, *item* and *alternation frequencies* are not manipulated independently of one another (*Wang et al., 2017*; *Higashi et al., 2017*) and, in some others, systematic analyses are not reported (*Tueting et al., 1970*); thereby precluding strong conclusions about the inference of *transition probabilities*. Here, by manipulating distinct dimensions of the space of *transition probabilities* (i.e. *item frequency* and *alternation frequency*) within the same experiment, we were able to demonstrate that mid-latency (~250 ms) and late (~500 ms) brain responses are sensitive to both *item* and *alternation frequencies* in a way that is best predicted by the inference of *transition probabilities*. Future studies whose goal will be to investigate *transition probability* learning should therefore systematically manipulate both *item* and *alternation frequencies* orthogonally within subjects.

Several computational models relying on predictive coding have been proposed to account for the effect of global and local *item frequency*, using (1) information theory (*Mars et al., 2008*), (2) the free-energy principle (*Lieder et al., 2013a*; *Lieder et al., 2013b*) or (3) constraints on the complexity of the representation used to generate predictions (*Rubin et al., 2016*). Our proposed modeling approach is in line with each of these influential proposals. As in (1), we show a correlation between brain responses and an information-theoretic measure (Shannon surprise). As in (2) we compute surprise using Bayesian inference. As in (3), we find that brain responses are well explained by the use of very recent observations to predict the upcoming one, and an attempt to summarize this recent history into a simple representation (here, we used *transition probabilities* between successive item, but we did not explore more complex possibilities).

Previous modeling studies often focused on a single brainwave, either the mismatch negativity (*Lieder et al., 2013a*; *Lieder et al., 2013b*) or the P300 component (*Squires et al., 1976*; *Mars et al., 2008*; *Kolossa et al., 2012*). Here, however, we conducted a systematic search across all post-stimulus latencies and sensors in order to identify brain responses that are modulated by various statistical properties of the input sequence. Using a similar approach, recent studies also showed that different computational aspects of learning could be mapped onto different electrophysiological signals (*Sedley et al., 2016*; *Diaconescu et al., 2017*). Here, this systematic approach combined with qualitative and quantitative model comparison (*Forstmann and Wagenmakers, 2015*; *Palminteri et al., 2017*) allowed the identification of a succession of computational processes in the brain. Early responses were best explained by the learning of the *frequency of items* over a

long timescale, while late responses were best explained by the learning of *transition probabilities* over an increasingly shorter timescale.

## Characterising timescales of integration

We modeled these timescales of integration using 'leaky' integrators, which have been shown to be biologically plausible and are therefore widely used in the modeling of many brain functions (*Rescorla and Wagner, 1972*; *Glaze et al., 2015*; *Farashahi et al., 2017*). We should, however, acknowledge a limitation of the present study: since we did not vary the timing between stimuli (unlike, for instance, *Pegado et al., 2010*), the timescale of integration here confounds elapsed time and the number of items, which cannot be separated by our experimental design.

We found that both long-term and short-term sequence learning mechanisms coexist in the brain, at different moments in the course of stimulus processing, with the timescale of integration decreasing progressively with response latency. Thus, previous studies that focused only on short-term (*Huettel et al., 2002*) or longer term (*Näätänen et al., 2007*) regularities may have missed some brain responses related to sequence learning. Our finding is, however, much in line with other studies suggesting that the brain may entertain several statistical estimates computed over different timescales (*Ulanovsky et al., 2004*; *Kiebel et al., 2008*; *Bernacchia et al., 2011*; *Ossmy et al., 2013*; *Meder et al., 2017*; *Scott et al., 2017*; *Runyan et al., 2017*). These multiple estimates might have an important computational role in the sense that they would allow the brain to detect regularities occurring at many different timescales while keeping the computations fairly easy and tractable, that is without relying on hierarchical Bayesian inference (*Collins and Koechlin, 2012*; *Wilson et al., 2013*).

Altogether, the different types of statistics that are learnt and the different timescales of integration that we identified in this study suggest an interesting dissociation, which has already been put forward in the past (*May and Tiitinen, 2010*; *Todorovic and de Lange, 2012*; *Strauss et al., 2015*; *Collins and Frank, 2018*), regarding the neural mechanisms at stake: early responses would reflect a habituation mechanism while later responses would be best explained by a predictive coding mechanism. In the first case, stronger signals to rare events come from the fact that neurons tuned to properties of rare items gets less habituated (i.e. they fatigue less) than those tuned to properties of the frequent items. In the second case, stronger signals evoked by rare events follow from violated expectations. In line with this dichotomy, we found that early responses are better explained by sensitivity to *item frequency* only, computed over a long history of stimuli. This has two consequences: first, only rare items (not rare repetitions or alternations) evoke strong signals at these latencies; second, this occurs at a very long timescale. Both aspects have been hypothesized as being cornerstones of habituation processes (*Kandel and Tauc, 1965*). On the other hand, mid-latency and late responses reflect a sensitivity to *transition probabilities*, computed over a much less protracted history. This translates into the fact that not only rare items but also rare transitions (be them alternations or repetitions) evoke stronger signals at these latencies. The simple fact that pairs of items, whose *item frequency* is matched, modulate brain signals at those latencies preclude an interpretation based on habituation mechanisms and are therefore only compatible with predictive coding (*Wacongne et al., 2012*).

## Multiple processings: from transition probabilities to chunks and patterns

Our findings also support the existence of multiple computational systems for sequence learning (*Dehaene et al., 2015*). The mid-latency response, with an integration characterized by a half-life of ~9 items, is compatible with a neuro-computational model of predictive coding, according to which the brain uses a window over the recent past in order to predict the *transition probability* from the current to the next item (*Wacongne et al., 2012*). The late response, on the other hand, is compatible with a conscious search for patterns (*Bekinschtein et al., 2009*; *Dehaene et al., 2015*; *Wang et al., 2015*) as shown by its extreme sensitivity to very recent observations (a half-life of ~4 items). The sensitivity to local *transition probabilities* identified here may be a mere consequence of this search for local patterns. We are indeed not claiming that the late brain signals we measured reflect solely *transition probability* learning. In fact, several studies have shown that late brain responses like the P300 reflect complex sequential dependencies (*Donchin and Coles, 1988*) such

as short melodic patterns (*Bekinschtein et al., 2009*; *Wacongne et al., 2011*; *Strauss et al., 2015*) and their abstract numerical and grammatical organization (*Wang et al., 2015*). Local *transition probability* learning, as modeled here, is probably just a minimal statistical approximation of what the brain systems generating those responses genuinely estimate.

One of such studies, the so-called 'local-global' paradigm (*Bekinschtein et al., 2009*; *Wacongne et al., 2011*), argues in favor of such a dissociation between brain systems characterized by an early latency which learn very simple sequential properties (like the *item frequency* or *alternation frequency*), and brain systems characterized by a late latency that track more complex sequential dependencies involving patterns of stimuli. In this paradigm, sequences are structured into chunks of five tones separated by a pause. Each chunk is a pattern which can be a series of repeated items (XXXXX) or a series of four repeated items followed by a deviant (XXXXY). In a modified version (*Strauss et al., 2015*), those patterns could also be a series of alternation (XYXYX) and a series followed by a deviant (XYXYY). In different experimental conditions, either pattern can itself be presented frequently (80% of patterns), thereby making the other pattern a rare deviant (20% of patterns). This experimental design thus manipulates independently within-chunk deviance (e.g. XXXXY and XYXYY) and between-chunk deviance (rare pattern). In this paradigm, early and mid-latency responses (the mismatch negativity, occuring at ~250 ms) were found sensitive to the *item frequency* (original version) and the *alternation frequency* (modified version): there is increased activity for the last item (Y) in both the XXXXY pattern and the XYXYY pattern. This first finding is fully compatible with the present observations and the hypothesis of a brain system learning *transition probabilities* because *item frequency* and *alternation frequency* are embedded in the space of *transition probabilities*. Furthermore, this increased activity for the last item (Y) is actually larger when the XXXXX pattern is more frequent (thus inducing a very low global frequency of Ys) compared to when the XXXXY pattern is more frequent (inducing a larger global frequency of Ys), suggesting that the corresponding timescale of integration is longer than five elements, compatible with the one we found in the present study for mid-latency responses (a half-life of ~9 items for the component occurring at ~250 ms).

A second finding from the 'local-global' paradigm is that late brain responses (the P300) are sensitive to the rarity of patterns over the entire block. While this could seem to contradict our present result that late brain responses are characterized by a very short timescale (a half-life of ~4 items for the component occurring at ~500 ms), they can be reconciled if one keeps in mind the critical difference between continuous sequences (the present study) and sequences structured into chunks (like the 'local-global' paradigm). With this distinction in mind, our current results and previous ones indeed support altogether the same hypothesis: late brain responses correspond to a brain system that operates over different levels of abstraction and within the space of working memory. This system would first process the (4 or 5) recent items and search for regularities among them. This system would then abstract those regularities in the form of a pattern, especially when the sequence is already segmented into chunks, such as 'the current pattern is made of alternating notes', or 'the current pattern comprises four identical notes followed by a different one'. Those patterns would serve as a new unit of processing: the brain would keep track of the patterns recently encountered and be surprised when an observation violates the pattern that recently prevailed. Such characteristics are typical of a rule-based search in conscious working memory, thus limited to a small number of elements (*Baddeley, 1992*). If this hypothesis is true, we can speculate that late brain responses observed here in the context of our prediction task will not be observed if participants are not attending to the sequence of stimuli. Those late brain waves are typically not observed in patients with impaired consciousness (*Faugeras et al., 2012*), during sleep (*Strauss et al., 2015*), or if the subject in not attending (*Bekinschtein et al., 2009*; *Chennu et al., 2013*). We used a prediction task here to maximize our chance of evoking late responses, and to be able to compare them with previous reports (*Squires et al., 1976*). On the contrary, earlier brain waves may arise from brain processes that automatically extract environmental statistics independently of task-relevance (*Bekinschtein et al., 2009*; *Wacongne et al., 2011*; *Strauss et al., 2015*). Testing the automaticity and task-dependence of our results will require further experiments.

Finally, while the present modeling effort was limited to the modeling of *transition probability* learning, it is likely that such learning constitutes only a building block on top of which higher-order representations are built, including the hierarchical structures that are thought to underpin language processing (*Saffran et al., 1996*; *Dehaene et al., 2015*; *Leonard et al., 2015*). Future studies will be

needed to better understand whether and how lower-order properties like *transition probabilities* can serve as a scaffold for more complex representation. A practical implication of our results is that such future studies should carefully control for possible biases in *transition probabilities*, otherwise effects attributed to higher-order representation could actually be driven by simple statistical biases.

## Materials and methods

### Subjects

Twenty participants (11 females), aged between 18 and 25 (mean 21.24, s.e.m.: 0.44) were recruited for this study. The study was approved by the local ethics committee (CPP 08–021 Ile-de-France VII), and participants gave their informed written consent before participating. Data from two subjects was discarded because of excessive head motion.

### Stimuli

Auditory and visual stimulation was delivered via Psychtoolbox (*Brainard, 1997*) running on MATLAB (Mathworks). The experiment was split into four different conditions. In each of these conditions, two different sounds were presented. These sounds were French syllables and were reported as being perceived without ambiguity by the subjects. The pair of syllables was changed at the beginning of each block: /ka/ and /pi/ in the first run, /ky/ and /te/ in the second one, /ku/ and /tɛ/ in the third one, and /pø/ and /to/ in the fourth one. Each syllable was pseudo-randomly labelled as A or B and lasted about 200 milliseconds, with a SOA between 2 syllables of 1.4 s (see *Figure 1A*). A fixation dot was displayed at the center of the screen while syllables were presented. To ensure subjects' attentional focus on the auditory stimulation, they were occasionally asked (every 12 to 18 sounds) to report an explicit discrete prediction (i.e. A or B) about the forthcoming stimulus using one of two buttons. Each block presented 400 to 409 stimuli and lasted for about 10 min. Subjects were allowed to rest in between.

### Experimental design

The four different conditions were characterized by different first-order *transition probabilities* (see *Figure 1B*): a condition with a frequency bias (i.e. $p(A|B) = 1/3$ and $p(B|A) = 2/3$ which led to $p(A) = 1/3$ and $p(alt.) = 4/9$), a condition with a repetition bias (i.e. $p(A|B) = p(B|A) = 1/3$ which led to $p(A) = 1/2$ and $p(alt.) = 1/3$), a condition with an alternation bias (i.e. $p(A|B) = p(B|A) = 2/3$ which led to $p(A) = 1/2$ and $p(alt.) = 2/3$) and a fully stochastic condition devoted of any bias (i.e. $p(A|B) = p(B|A) = 1/2$ which led to $p(A) = 1/2$ and $p(alt.) = 1/2$). The order of conditions was pseudo-randomized across subjects.

### Data acquisition

Subjects' brain activity was recorded using a 306 channels (102 triplets of sensors each composed of 1 magnetometer and two orthogonal planar gradiometers) whole-head Elekta Neuromag MEG system with an acquisition sampling rate of 1 kHz and hardware bandpass filtering between 0.1 and 330 Hz. Prior to installing the subjects in the MEG room, we digitized three head landmarks (nasion and pre-auricular points), four head position indicator (HPI) coils placed over frontal and mastoïdian skull areas, as well as about 60 additional locations over subjects' skull using a three-dimensional Fastrak system (Polhemus). Thanks to HPI landmarks, head position within the MEG helmet was measured at the beginning of each run. Electrocardiogram, vertical and horizontal oculogram were recorded in order to monitor heart rate and eye movements during the experiment.

### Pre-processing

Raw MEG signals were first corrected for between-session head movements and bad channels using MaxFilter (Elekta). Following pre-processing steps were performed using Fieldtrip in MATLAB (Mathworks). First, MEG data was epoched between −250 ms to 1 s with respect to sound onset. Power line artefacts were then removed by filtering out the 50 Hz frequency and its 100 and 150 Hz harmonics. Trials exhibiting muscle and/or other movements' artefacts and/or SQUIDs' jumps that were detected based on semi-automatic methods (using the variance of MEG signals over sensors and the first order derivative of MEG signals over time) were then completely removed from the

dataset. MEG signals were then low-pass filtered below 30 Hz and downsampled to 250 Hz. An ICA was then performed and components exhibiting topographical and time courses signatures of eye blinks or cardiac artefacts were subsequently removed from the data. The remaining trials were eventually monitored visually and individually to ensure that all bad trials were removed from the subject's dataset. Last, data were baseline-corrected by removing trial-wise averaged activity in the 250 ms time window preceding sound onset. Throughout the paper results are reported only for magnetometers.

## Sources localizations

Anatomical T1-weighted MRIs (with $1 \times 1 \times 1.1$ mm voxel size) were acquired for each subject using a 3T Tim Trio (Siemens) MRI scanner. For each subject, grey matter was first segmented from white matter using FreeSurfer (*Fischl et al., 1999*). Segmented tissues were then imported into BrainStorm (*Tadel et al., 2011*) in order to reconstruct subjects' head and cortical surfaces using a cortical mesh composed of 150,002 vertices. Models of the cortex and the head were used to estimate the current-source density distribution over the cortical surface. To do so, subject-specific noise covariance matrices were estimated in each condition on the baseline from the 250 ms time window preceding sound onset. Individual sources were computed using weighted minimum-norm estimates (depth weighting factor: 0.8, losing factor for dipole orientation: 0.2), separately for each condition, and the norm of each tridimensional vector obtained was finally computed for each vertex. The resulting sources were projected on a standard anatomical template and averaged across subjects. The group average was finally interpolated to a higher resolution brain template composed of 306716 vertices that was inflated to 35% to ensure high-quality figures (see *Figure 1—figure supplement 2B*).

## Statistical analyses

We searched for differences between rare and frequent observations within each experimental condition. To correct for multiple comparison over sensors and time points, we used spatiotemporal clusters and non-parametric paired *t*-tests estimated from 1000 permutations (*Maris and Oostenveld, 2007*). A cluster was defined by adjacent time points and neighbouring sensors (neighbours sensors are less than 15 cm apart; there is an average of 8.3 neighbours per sensor). The cluster-level statistic was the sum of the sample-specific *t*-statistics that belong to a given cluster. The alpha level of the sample-specific test statistic was 0.05 and the alpha level of the cluster-specific test statistic was 0.05.

## Topographical filtering

Group averaged topographical 'surprise' responses (see *Figure 3—figure supplement 1*) were computed by contrasting, in a late time window (from 500 to 730 ms), MEG signals evoked by local streaks that were violated (e.g. XXX<u>Y</u>), both given local (e.g. a local series of repetitions) and global (e.g. alternations are globally less frequent than alternations in the alternation-biased condition) statistics, against the same streaks that were continued (e.g. XXX<u>X</u>). Those group-level topographies were then used to filter, at the subject level, MEG signals obtained in the same time window and the same sequence, but evoked by all the possible patterns up to length 4 (see *Figure 3*). Importantly, we used a cross-validation approach to ensure statistical validity and preclude double dipping (*Kriegeskorte et al., 2009*). More precisely, topographical filters were estimated at the group level on *odd* trials and applied at the subject level on *even* trials. We also did it the other way around and pooled both together. In order to facilitate visual comparison, in each condition, the averaged activity was subtracted from the activity evoked by each pattern (see *Figure 4*).

## Bayesian models of the learning process

We designed 'ideal observers' that infer the hidden value of a statistic $\theta$ of stimuli given a particular sequence $u$ of binary observations. For the sake of simplicity, we first consider here the learning of IF ($M_{\text{IF}}$). This ideal observer infers the frequency of As ($\theta^{\text{A}}$), and updates it following each observation, in an optimal manner following Bayes' rule.

$$p\left(\theta^{\text{A}}|u_{1:k}\right) = \frac{p\left(u_{1:k}|\theta^{\text{A}}\right) \cdot p\left(\theta^{\text{A}}\right)}{p\left(u_{1:k}\right)}$$

Because observations are binary and our prior about $\theta^A$ is uniform, the posterior distribution follows a Beta distribution (*Gelman et al., 2013*).

$$p\left(\theta^A | u_{1:k}\right) \propto \mathrm{Beta}\left(N_k^A + 1, N_k^B + 1\right)$$

The mean of this distribution, which is also the predictive likelihood that the next observation will be an A, can be expressed analytically as follows.

$$\mathbb{E}\left(\theta_k^A\right) = \frac{N_k^A + 1}{N_k^A + N_k^B + 2} = p\left(u_{k+1} = A | u_{1:k}, \mathcal{M}_{IF}\right)$$

Where $N^A$ and $N^B$ denote the number of A and B items that have been observed up to the $k$-th observation of the sequence.

Following *Shannon (1948)*, the theoretical surprise levels can then be computed as the predictive likelihood of the observation that is actually received.

$$I_{k+1} = \begin{cases} -\log_2\left(\mathbb{E}\left(\theta_k^A\right)\right) & \text{if } u_{k+1} = A \\ -\log_2\left(1 - \mathbb{E}\left(\theta_k^A\right)\right) & \text{if } u_{k+1} = B \end{cases}$$

Because integration over long periods of time and a large number of observations is often compromised by memory constraints, we designed models with different timescales of integration. In those models, the weight of a given observation decays exponentially with the number of observations. The count of observations $N^A$ and $N^B$ is therefore 'leaky'. For mathematical convenience, let us assume that the sequence of observation $u_{1:n}$ is coded numerically as 0 s and 1 s representing respectively Bs and As. In that case, the leaky count of the occurrences of item A is:

$$N_\omega^A = \sum_{k=1}^{n} u_{n-k} - \exp\left(\frac{-k}{\omega}\right)$$

Where $\omega$ is a free parameter that characterizes the strength of the leak and therefore controls the timescale of integration (see *Figure 5—figure supplement 2*): smaller values of $\omega$ correspond to a more local integration that favours more recent observations. When $\omega = \infty$, the integration becomes perfect, that is devoid of any forgetting, and thus provides an optimal estimate of the true generative statistics that are fixed within each condition of our experiment. Importantly, the timescale of integration determines the stability of beliefs, and therefore influences the dynamics of surprise levels (see *Figure 2—figure supplement 1B* for an example): the smaller the $\omega$, the more local the integration and the more erratic the surprise levels.

Now that we have presented the derivation for the ideal observer learning the *frequency of items* IF, we can extend it to other types of observers that learn AF or TP. The derivation for the AF model is most similar to the IF model: while the IF model counts, with a leak, the occurrence of As (and Bs), the AF model counts, with a leak, the occurrence of alternations (and repetitions). For the AF model, one therefore simply needs to recode the input sequence into repetitions and alternations (rather than As and Bs) and apply the exact same logic as for the IF model. The derivation of the TP model is slightly different, since it requires to estimate two statistics (instead of one, in IF and AF): two *transition probabilities*, one for the probability of observing an A after a B $\theta^{A|B} = 1 - \theta^{B|B}$ and one for the probability of observing a B after an A, $\theta^{B|A} = 1 - \theta^{A|A}$. Note that $\theta^{B|B}$ and $\theta^{A|A}$ can be deduced from $\theta^{A|B}$ and $\theta^{B|A}$, such that two probabilities suffice to describe all transitions. Each TP is estimated essentially like the *frequency of items*, but by counting transitions. The estimation of $\theta^{A|B}$ and $\theta^{B|A}$ therefore requires to entertain (leaky) counts corresponding to all possible transitions (AA, AB, BA, BB). More details are provided in *Meyniel et al. (2016)* and the MATLAB code for all ideal observers is available (*Meyniel and Maheu, 2018*; copy archived at https://github.com/elifesciences-publications/MinimalTransitionProbsModel).

It is important to notice that IF and AF can be analytically derived from TP:

$$\theta^A = \frac{\theta^{A|B}}{\theta^{A|B} + \theta^{B|A}} \text{ and } \theta^{\text{alt.}} = 2 \cdot \frac{\theta^{A|B} \cdot \theta^{B|A}}{\theta^{A|B} + \theta^{B|A}}$$

Thus, IF and AF are subsumed in the space of TP (see *Figure 1A*).

## Trial-by-trial regressions

Following previous work (*Mars et al., 2008*), we hypothesize that, for each trial $k$, brain signals $y_k$ linearly scale with the theoretical level of surprise $I_k$. Here, different ideal observer models $M$ learning a different statistics $i$ with a difference timescale of integration $\omega$ have different surprise levels. We thus estimated regression coefficients $\beta$ from the following linear regression model (see *Figure 5*).

$$y_k = \beta_0^b + \beta_1 \cdot I_k(\mathcal{M}_i, \omega) + \epsilon \text{ with } \epsilon \sim \mathcal{N}\left(0, \sigma_\epsilon^2\right)$$

We estimated regressions for each subject, at each time point around sound onset and each MEG sensor. We also varied, in separate regressions, the ideal observer model (IF, AF, TP) and the timescale of integration $\omega$. We tested 55 values of $\omega$ (see *Figure 5E* for the exact grid used) so as to cover the entire number of observations in the sequence (including the case of a perfect integration: $\omega = \infty$) while minimizing correlations between surprise levels obtained with different values of $\omega$.

## Bayesian model comparison

In order to compare models, we adopted a Bayesian approach. We approximated the marginal likelihood (or model evidence) of each model, a key metric for model comparison, using the Bayesian information criterion (BIC; *Schwarz, 1978*). Under independent and identically distributed model errors, the BIC can be expressed as follows.

$$\text{BIC} = n \cdot \log \hat{\sigma}_\epsilon^2 + k \cdot \log n$$

$$\hat{\sigma}_\epsilon^2 = \min \frac{1}{n} \cdot \sum_{k-1}^{n} \left(y_k - \hat{y}_{k,\omega}\right)^2$$

where $n$ is the number of observations, $\kappa$ the number parameters: that is here 4 $\beta_0$s (one per condition), $\beta_1$ and $\omega$. Note that all models here have the same number of free parameters. Therefore, other approximations for the model evidence such as the Akaike Information Criterion (*Akaike, 1998*), or simply the proportion of explained variance, would lead to the same conclusions.

We then used model evidence to compare models with a random-effect approach. An alternative is to use a fixed-effect approach, by summing log-ratios of model evidence (i.e. Bayes Factors). However, such an approach is valid only for pairs of models and its results can easily be driven by outlier subjects. We performed the random-effect analysis described in *Stephan et al. (2009)*; *Rigoux et al. (2014)* and implemented in the VBA toolbox (*Daunizeau et al., 2014*). This analysis returns several key estimates for each model included in the comparison set: the expected model frequency in the general population as well as the probability for each model to be more frequent than any other model in the general population (the 'exceedance probability', noted $\varphi$).

## Bayesian model averaging

In addition to identifying the model that best explain the MEG signals, one can also draw inference about the best integration parameter $\omega$. To do so, the BIC approximation of the model evidence is estimated for each value of the timescale integration parameter $\omega$.

$$p(y|\mathcal{M}, \omega) \approx \exp\left(\frac{-\text{BIC}(\omega)}{2}\right)$$

Note that the equation above depends on the use of a particular model. Since we do not know for sure which model is used by the brain, we can combine all models following probability calculus in order to infer the value of $\omega$ that best explain the MEG signals. Under a uniform prior over $M$ and $\omega$, it boils down to:

$$p(\omega|y) \propto \sum_i p(y|\mathcal{M}_i, \omega)$$

Note that those equations apply to one subject, such that we obtain an estimate of $\omega$ per subject.

## Acknowledgements

We thank all the volunteers for their participation. We thank Sébastien Marti, Emmanuel Procyk and Valentin Wyart for useful discussions throughout the project. We are grateful to the nurses Véronique Joly-Testault, Laurence Laurier, and Gaëlle Médiouni for their help in recruiting subjects and data acquisition.

## Additional information

### Funding

| Funder | Grant reference number | Author |
|---|---|---|
| Fondation Bettencourt Schueller | "Frontières du Vivant" doctoral fellowship | Maxime Maheu |
| Ministère de l'Enseignement supérieur, de la Recherche et de l'Innovation | "Frontières du Vivant" doctoral fellowship | Maxime Maheu |
| Institut National de la Santé et de la Recherche Médicale | | Stanislas Dehaene |
| Commissariat à l'Énergie Atomique et aux Énergies Alternatives | | Stanislas Dehaene Florent Meyniel |
| Collège de France | | Stanislas Dehaene |
| European Research Council | Grant NeuroSyntax | Stanislas Dehaene |
| European Union Seventh Framework Programme | 604102 - Human Brain Project | Stanislas Dehaene Florent Meyniel |

The funders had no role in study design, data collection and interpretation, or the decision to submit the work for publication.

### Author contributions

Maxime Maheu, Data curation, Formal analysis, Validation, Investigation, Visualization, Methodology, Writing—original draft; Stanislas Dehaene, Conceptualization, Supervision, Funding acquisition, Validation, Writing—review and editing; Florent Meyniel, Conceptualization, Supervision, Validation, Writing—review and editing

### Author ORCIDs

Maxime Maheu  http://orcid.org/0000-0002-6851-4927
Florent Meyniel  https://orcid.org/0000-0002-6992-678X

### Ethics

Human subjects: The study was approved by the local ethics committee (CPP 08-021 Ile-de-France VII), and participants gave their informed written consent before participating.

### Decision letter and Author response

Decision letter https://doi.org/10.7554/eLife.41541.022
Author response https://doi.org/10.7554/eLife.41541.023

## Additional files

### Supplementary files

• Transparent reporting form
DOI: https://doi.org/10.7554/eLife.41541.017

## Data availability

The preprocessed MEG data are available via the Open Science Framework (https://osf.io/wtnke/). MATLAB code of the (ideal) Bayesian observers can be found at https://github.com/florentmeyniel/MinimalTransitionProbsModel (copy archived at https://github.com/elifesciences-publications/MinimalTransitionProbsModel).

The following dataset was generated:

| Author(s) | Year | Dataset title | Dataset URL | Database and Identifier |
|---|---|---|---|---|
| Maxime Maheu | 2019 | Brain signatures of a multiscale process of sequence learning in humans | https://osf.io/wtnke/ | Open Science Framework, wtnke |

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
