## [Decision Letter]

Thank you for submitting your article "Brain Signatures of a Multiscale Process of Sequence Learning in Humans" for consideration by *eLife*. Your article has been reviewed by three peer reviewers, including Floris de Lange as the Reviewing Editor and Reviewer #1, and the evaluation has been overseen by Timothy Behrens as the Senior Editor. The following individual involved in review of your submission has agreed to reveal their identity: Marta Garrido (Reviewer #3).

The reviewers have discussed the reviews with one another and the Reviewing Editor has drafted this decision to help you prepare a revised submission.

Summary:

Maheu et al. perform a systematic and thorough investigation of the effects of item frequency (IF), alternation frequency (AF) and transition probability (TP) of auditory events on the MEG response in different time windows. They find that IF, estimated over long temporal windows, explains the brain response in very early auditory sensory processing, whereas TP, estimated over progressively shorter time windows, explains the brain response at progressively later time points.

Essential revisions:

1) Task-dependency of results?

The participants were explicitly required to judge the predictability of the patterns. It is therefore very likely that the observed brain responses (especially the later ones) reflect this perceptual state and therefore cannot be interpreted as automatic processes reflecting the brain's modus operandi as a statistical learning machine. It would have been far better, if the task performed by participants was unrelated to the sequence dimension being investigated, e.g. some sort of omission detection task? Or an entirely unrelated (e.g. visual) task. The present choice limits the interpretation of the results. It will be important to better qualify what inferences can and cannot be drawn on the basis of these data.

Related, the major findings that (1) brain responses reflect the statistics of sound sequences (2) early responses reflect adaptation-like effects whereas later effects reflect increasingly complex models is not new. The authors themselves review lots of relevant literature. Notably, they fail to relate their results to the modelling work of Rubin et al., 2006. Though the paper is cited in the present report in the context of the claim that brain responses are larger for deviants than standards, the actual work has employed extensive modelling to understand how neural responses to sequences similar to those used here are affected by the preceding history. Notably, they have concluded that responses are affected by quite a lengthy context (more than 10 tones) but that the representation of the transition probabilities is coarse. Is it possible that the different effects observed here are driven by the behavioral task (see my comment no. 1, above)?

2) Choice of time window:

The spatial filter analysis is focused on the late time window. The one that is most contaminated by the behavioral task. What is the justification of using that time window and not and earlier one?

In previous and later figures, IF and AF seem to be doing all the work at much earlier time windows.

3) Various methodological concerns: a) Figure 5 the time course of max R2 nicely aligns with the three MEG peaks. Could it be that the appearance of "3 distinct time windows" is just a consequence of MEG power?

b) Figure 2 shows significant differences between conditions (B vs. A, and Rep vs. Alt) overlaid on MEG topography for the 4 different sequence conditions. While it is interesting to observe the differences in topography for the MEG difference wave across sequence conditions, this remains a qualitative observation. Statistical maps would be required to really pin down whether those patterns are indeed significantly different between conditions, and if yes, where in space-time. For example Figure 2B, it appears that from 400 ms onwards the contrast of Rep vs. Alt is a little different between fully stochastic and repetition biased sequences, but, is this difference (interaction) really significant?

c) Figure 4 shows a remarkable similarity between empirical MEG data and the surprise levels predicted by the transition probability model for 500-730 ms, which is very impressive and pleasing. It was unclear, however, whether this time window was a priori chosen or whether the whole window was fully explored with appropriate multiple testing correction. If a priori chosen then more explanation for the rationale is needed given that a) strong evidence exists for MMN window 100-250 ms and b) Figure 3A shows significant differences for earlier periods – in fact for the alternation biased condition no significant differences between rep and alt exist beyond 400 ms so that's difficult to reconcile with the later time window. If based on Figure 5D then why not from 160ms from which point TP wins?

d) Still related to the findings in Figure 4 that related to a local integration window parameterised by ω =6. My understanding is that this parameter was optimised for each subject and that 55 values (as many as the max of observations made) were explored. How consistent was this value across participants (ω =16 is also mentioned)? It appears this was also optimised for each of the 3 times windows – can this be made more explicit and also again whether that was consistent over subjects. Did it correlate with their task accuracy (more accurate prediction judgments on upcoming stimuli for greater ω)? This is not necessary but could strengthen the interpretation.

e) The fact that global statistic violations such as item frequency violation modulates earlier (not late) responses matches nicely Garrido 2013 narrow/broad Gaussian paper showing that variance (a global statistic) modulates early latency responses. However, it appears at odds with Bekinschtein's 2009 local-global paper showing that global violations modulate P300 (not MMN). Can we reconcile these findings? The Discussion appears to point to chunk vs. continuous sequences as potentially explaining the divergence (which would again be consistent with Garrido 2013), but it still remains unclear why would chunking make such a difference?

---

## [Author Response]

Essential revisions:1) Task-dependency of results?The participants were explicitly required to judge the predictability of the patterns. It is therefore very likely that the observed brain responses (especially the later ones) reflect this perceptual state and therefore cannot be interpreted as automatic processes reflecting the brain's modus operandi as a statistical learning machine.

We completely agree with the reviewers that some of the observed brain responses, in particular late ones, may depend upon the fact that the sequence of stimulus is task-relevant, as previously reported by e.g. Pitts et al., 2014, and Strauss et al., 2015. However, it was not our intention to suggest that our results reflect an automatic and unconscious extraction of statistical regularities. We have screened our text searching such a misleading interpretation, but could not find any. We are happy to take suggestions. In order to clarify this point, we now add a paragraph in the Discussion, indicating that several aspects of our conclusions may be task-dependent, and that it is unclear whether the same effects would be observed if the subject were engaged in a distracting task, not paying attention to stimuli, or even asleep (see next point).

It would have been far better, if the task performed by participants was unrelated to the sequence dimension being investigated, e.g. some sort of omission detection task? Or an entirely unrelated (e.g. visual) task. The present choice limits the interpretation of the results. It will be important to better qualify what inferences can and cannot be drawn on the basis of these data.

The fact that the behavioural task was related to the sequence dimension was actually a deliberate choice. We wanted to maximize the number and variety of brain processes we would record. In particular, we wanted to relate our study to different previous studies, including those on P300/slow-wave components, which are rarely seen for unattended or irrelevant stimuli. Indeed, an important goal of our study was to extend the seminal results by Squires et al., 1976, who reported a modulation of the P300/slow-wave EEG component by the local and global item frequency. We did so by including in our study a new computational model as well as new experimental conditions (with repetition and alternation biases). In Squires et al.’s 1976 study, subjects counted the high pitch tone, so that the stimulus was task-relevant. Using an entirely unrelated task would have jeopardized the study of late brain responses and comparability with previous studies. To avoid possible misunderstandings, we have added a new paragraph to the Discussion in order to clarify this point:

“[…] we can speculate that late brain responses observed here in the context of our prediction task will not be observed if participants are not attending to the sequence of stimuli. Those late brain waves are typically not observed in patients with impaired consciousness (Faugeras et al., 2012), during sleep (Strauss et al., 2015), or if the subject in not attending (Chennu et al., 2013). […] Testing the automaticity and task-dependence of our results will require further experiments.”

Related, the major findings that (1) brain responses reflect the statistics of sound sequences (2) early responses reflect adaptation-like effects whereas later effects reflect increasingly complex models is not new. The authors themselves review lots of relevant literature. Notably, they fail to relate their results to the modelling work of Rubin et al., 2006. Though the paper is cited in the present report in the context of the claim that brain responses are larger for deviants than standards, the actual work has employed extensive modelling to understand how neural responses to sequences similar to those used here are affected by the preceding history. Notably, they have concluded that responses are affected by quite a lengthy context (more than 10 tones) but that the representation of the transition probabilities is coarse. Is it possible that the different effects observed here are driven by the behavioral task (see my comment no. 1, above)?

The modelling and experimental work that we quoted by Rubin et al., 2016,) is indeed highly relevant to our work (we assumed that the reviewer mistyped 2006 instead of 2016). Their theoretical account proposes a new way of predicting future observations while constraining the complexity of the representation of the past that is used for the prediction. They tested the model’s predictions on data obtained in an oddball task, and show that the model indeed predicts higher activity for the deviant tones (rare items) compared to the standard ones (frequent items). They also show that their model is crucially characterized by a local inference and a coarse representation of the past.

Important differences with this prior work should nevertheless be stressed: it presents single-unit recordings in anesthetized cats (while we have MEG data in behaving humans), it uses only two of our four experimental conditions (the frequency-bias condition and the no-bias condition), it does not explore the dynamics of the stimulus-evoked response. The model is also different: we restricted ourselves to transition probabilities of order 1 while Rubin et al., 2016, go to order 50. Our choice was motivated by our experimental design, which is tailored for testing the order 1 with the inclusion of a repetition-bias and alternation-bias condition, and the fact that investigating higher order is difficult due to combinatorial explosion and the limited amount of data available in typical experiments (sequences have 430 stimuli in our experiment). Differences between our study and Rubin et al., 2016, should therefore be interpreted with care.

Despite those differences, there are two major points of convergence between Rubin et al., 2016, and our study. First, prediction relies on a local inference: Rubin et al., 2016, found that a history of more than 10 items is used; we found an exponential decay with ω = 13 and 6 for the intermediate and late time window; and we found ω = 16 for Squires et al.’s 1976 dataset. Second, in order to make predictions, the brain uses a compressed representation of the past; in our case, it is in the form of transition probabilities; in Rubin et al., 2016, their explore more complex representations.

In the revised version, we have added a new paragraph in the Discussion, see below) that lists the influential theoretical accounts that have been proposed regarding the modeling of brain responses to sequences characterized by a bias in the frequency of items (e.g. oddball sequences), and in particular, discuss more the study by Rubin et al., 2016.

“Several computational models relying on predictive coding have been proposed to account for the effect of global and local item frequency, using (1) information theory (Mars et al., 2008), (2) the free-energy principle (Lieder et al., 2013a, 2013b) or (3) constraints on the complexity of the representation used to generate predictions (Rubin et al., 2016). […] As in (3), we find that brain responses are well explained by the use of very recent observations to predict the upcoming one, and an attempt to summarize this recent history into a simple representation (here, we use transition probabilities between successive item, but we did not explore more complex possibilities).”

2) Choice of time window:The spatial filter analysis is focused on the late time window. The one that is most contaminated by the behavioral task. What is the justification of using that time window and not and earlier one?In previous and later figures, IF and AF seem to be doing all the work at much earlier time windows.

First, we would like to clarify that none of our data are not contaminated by behavior: motor responses are restricted to ~1/15 of the trials, which are not included in the analysis. We added a sentence in the Materials and methods to clarify. If the reviewer is referring to a potential task-dependency of our results, please see our response above.

Second, the analysis that involves the spatial filter is intended as an extension of the seminal and influential paper by Squires et al., 1976, in which the authors reported a modulation of P300 and late slow-wave by the sequence statistics (both local and global) in an experiment that contained two of our four conditions. We therefore selected a similar late time-window, in order to allow comparison with this study and subsequent ones on the same topic (Bekinschtein et al., 2009; Donchin and Coles, 1988; Feuerriegel et al., 2018; Kolossa et al., 2012; Mars et al., 2008; Melloni et al., 2011; Meyniel, Maheu and Dehaene, 2016; Pegado et al., 2010; Squires et al., 1976; Strauss et al., 2015; Summerfield et al., 2011; Wacongne et al., 2011).

Third, note that while we provide a Squires-style, detailed analysis of the signal in an (arbitrary or “historical”) late time window, the remainder of our analysis (Figures 2, 5, Figure 2—figure supplement 1, Figure 2—figure supplement 2, Figure 2—source data 1 and even more in the new Figure 2—figure supplement 1 and Figure 5—figure supplement 1 that is also presented below) probes the MEG signal in a systematic manner, independently from any prior assumption regarding the timing of effects (which post-stimulus latency), their spatial location (which sensors) and their tuning properties (modulatory effect of local/global statistics, and of which statistics).

In the revised version, we have added a sentence in the Results section that explain our general approach and the fact that the late time window was chosen for historical reasons (subsection “Description of auditory-evoked responses”, last paragraph, see below).

3) Various methodological concerns:a) Figure 5 the time course of max R2 nicely aligns with the three MEG peaks. Could it be that the appearance of "3 distinct time windows" is just a consequence of MEG power?

Theoretically speaking, it could have been the case that our model explains the signal (*R*^2^) mostly when evoked responses are strong. However in practice, it is not the case: the (3) peaks in *R*^2^ do not really align with the (4) peaks observed for MEG (global field) power (see Author response image 1). In particular, the second and fourth peaks of the MEG power (blue) fall exactly in between two peaks of *R*^2^ (blue).

We have amended our Figure 5 so as to now include the global field power for reference.

b) Figure 2 shows significant differences between conditions (B vs. A, and Rep vs. Alt) overlaid on MEG topography for the 4 different sequence conditions. While it is interesting to observe the differences in topography for the MEG difference wave across sequence conditions, this remains a qualitative observation. Statistical maps would be required to really pin down whether those patterns are indeed significantly different between conditions, and if yes, where in space-time. For example Figure 2B, it appears that from 400 ms onwards the contrast of Rep vs. Alt is a little different between fully stochastic and repetition biased sequences, but, is this difference (interaction) really significant?

We should distinguish two types of comparisons: *within* and *between* conditions. Our statistical analysis, which is corrected for multiple comparisons over sensors and time points, addresses within-condition comparisons: Do A and B produce different signals when there exist a frequency bias? Do alternation and repetition produce different signals within the no-bias condition? And similarly, within the repetition-bias condition, and within the alternation-bias condition? We revised our text to avoid any ambiguity.

We nevertheless agree with the reviewers that the between-condition comparisons are also interesting, albeit different. We partially address between-conditions effects later in the paper, when we focus on (late) brain responses that reflected a sensitivity to *all* types of (local) statistics (i.e. in *all* conditions).

In order to assess it systematically, we now provide, as suggested by the reviewers, the statistical maps corresponding to pairwise differences between frequent – rare events across conditions.

As anticipated by the reviewers, the statistical analysis indeed reveals a significant difference between the Rep – Alt contrast between fully-stochastic and repetition-biased conditions starting around 500 ms which we included as a new Figure 2—figure supplement 2.

c) Figure 4 shows a remarkable similarity between empirical MEG data and the surprise levels predicted by the transition probability model for 500-730 ms, which is very impressive and pleasing. It was unclear, however, whether this time window was a priori chosen or whether the whole window was fully explored with appropriate multiple testing correction. If a priori chosen then more explanation for the rationale is needed given that a) strong evidence exists for MMN window 100-250 ms and b) Figure 3A shows significant differences for earlier periods – in fact for the alternation biased condition no significant differences between rep and alt exist beyond 400 ms so that's difficult to reconcile with the later time window. If based on Figure 5D then why not from 160ms from which point TP wins?

We thank the reviewers for raising this important question.

First, note that Figure 4 is an extension of Figure 3: we therefore used the same time-window. The choice of this late time window is motivated to allow comparison with the study by Squires et al., 1976, see point 2 above. This analysis relies on the use of spatial filters, the choice of the exact time window was done a priori based on the filters’ profiles (see Figure 3—figure supplement 1): we selected manually a single late time window in which all filters showed strong signals. This choice was not further optimized to show similarity with the local transition probability model (which would have required correction for multiple testing, as pointed by the reviewers).

Also note that Figure 4 is mostly for illustration purpose: to show that the local transition probability model *qualitatively* accounts for the late evoked responses which were of interest in previous studies (Bekinschtein et al., 2009; Donchin and Coles, 1988; Feuerriegel et al., 2018; Kolossa et al., 2012; Mars et al., 2008; Melloni et al., 2011; Meyniel, Maheu and Dehaene, 2016; Pegado et al., 2010; Squires et al., 1976; Strauss et al., 2015; Summerfield et al., 2011; Wacongne et al., 2011). A more quantitative and exhaustive analysis (for all time points and sensors) is provided in Figure 5.

We have added two sentences explaining the choice of that late time-window in the Results section (see below), and we amended Figure 3—figure supplement 1 with a new panel, in order to clarify that the time window is selected a priori based on the filters, and not post-hoc based on the cross-validated analyses reported in Figure 3 and Figure 4.

“Note that this late time window (from 500 to 730 ms) was selected in order to follow up and expand the seminal study of Squires et al., 1976, which focused on late (EEG) brain responses. Here, the exact time window was not optimized to show the clear tree structure seen in Figure 3, or the strong similarity between observed data and the model reported in Figure 4, but selected a priori (see Figure 3—figure supplement 1).”

Regarding the specific question about effects in the alternation-biased condition beyond 400 ms: Figure 2 shows no significant effect when correcting for multiple comparisons over time and sensors, but this is of course conservative and dependent on a *p*-value threshold, the topographies nevertheless show (below-threshold) effects. Those effects are pooled across sensors by the spatial filter in a time window in Figure 4, which further reveals the effect: there is no contradiction in the data. In actuality, the Alt. vs. Rep. effect is not significantly stronger in the alternation-bias than in the repetition-bias condition (see response to 3b).

*d) Still related to the findings in Figure 4 that related to a local integration window parameterised by* ω *=6. My understanding is that this parameter was optimised for each subject and that 55 values (as many as the max of observations made) were explored. How consistent was this value across participants (*ω *=16 is also mentioned)? It appears this was also optimised for each of the 3 times windows – can this be made more explicit and also again whether that was consistent over subjects.*

Figure 4 presents theoretical predictions, which can be computed only after parameterizing our model (ω, the time scale of integration, is the only free parameter). We have not optimized ω for Figure 4, actually we took the value that we identified in the exhaustive analysis presented in Figure 5 for the late time window (because Figure 4 is about late responses).

In order to obtain this value of *ɷ* in Figure 5, we computed for each subject the posterior distribution *p*(ω|MEG) which gives, for each of the 55 values of ωthat were tested (see vertical grid on Figure 5E), how likely that particular value of ω is given the observed MEG activity in that subject (see Materials and methods). This distribution is obtained through Bayesian Model Averaging (BMA) which takes into account the uncertainty at the level of model inference (IF, AF and TP). We then averaged over subjects, and reported the mean distribution in Figure 5E. The maximum a posteriori value was defined on the group-level distribution for each time window of interest. We rephrased the legend of Figure 5 and the main text to clarify.

“Finally, in order to draw inference about the timescale of integration of each of these three signatures, we performed a Bayesian Model Averaging (BMA, see Materials and methods). […] At the group level (see Figure 5E and Figure 5—figure supplement 2), the first window was best characterized by a global integration (ω = ∞), the second and third by much more local integration (ω = 13 and 6 respectively, maximum a posteriori values; i.e. a half-life of 9.01 and 4.16 observations respectively).”

Those estimates of ωshould be taken with caution. First, the posterior distributions in Figure 5 shows that actually the range of probable values is rather large. The reviewers mention ω= 16, which is the value that we found in our previous paper (Meyniel, Maheu and Dehaene, 2016) when fitting the dataset by Squires et al., 1976. We also found ω = 19 for another EEG dataset (Kolossa et al., 2013). All those values have a very high posterior probability in our current data (see Figure 5E). Differences between those values may also reflect that the evoked responses is taken at different (late) latencies and specific aspects of each data set. Second, the exact values for each participant are rather spread out, especially maximum a posteriori values. Mean values are more consistent, but this is artifactually due to a regression to the (average) prior value. This is why we report group-level estimates (rather than a mean of subject-level estimates) to achieve a more stable result. Note that while individual exact values are spread out, the decrease of those values from early to late latencies is very consistent across subjects (*t*_17_ = –3.00, *p* = 0.008, already reported in the text).

*Did it correlate with their task accuracy (more accurate prediction judgments on upcoming stimuli for greater* ω*)? This is not necessary but could strengthen the interpretation.*

The answer to this question depends on how accuracy is measured. An objective measure is to consider the average likelihood of subjects’ choices based on the generative transition probabilities. For instance, in the repetition-biased condition, reporting expecting a A after a A comes with a higher likelihood given the generative transition probabilities than expecting a B after the same A. With such a measure, subjects would indeed achieve a better performance if they were using a longer integration (i.e. greater ω). However, this measure of accuracy did not correlated significantly with the subject’s maximum a posteriori ω parameter (with mean ω: *ρ* = –0.11, *p* = 0.66; with maximum a posteriori ω: *ρ* = –0.16, *p* = 0.54). We are reluctant to include this non-significant result in the paper because our study is most likely underpowered for testing between-subject correlations (*N* = 18 subjects).

e) The fact that global statistic violations such as item frequency violation modulates earlier (not late) responses matches nicely Garrido 2013 narrow/broad Gaussian paper showing that variance (a global statistic) modulates early latency responses. However, it appears at odds with Bekinschtein's 2009 local-global paper showing that global violations modulate P300 (not MMN). Can we reconcile these findings? The Discussion appears to point to chunk vs. continuous sequences as potentially explaining the divergence (which would again be consistent with Garrido 2013), but it still remains unclear why would chunking make such a difference?

First, we would like to stress that late responses are also modulated by global statistics (see e.g. Figure 2A and Figure 4A), including the violation of item frequency.

Second, we acknowledge that terminology plays against us. In this paper, local/global refers to the timescale of integration (i.e. number of items processed) following previous publications (Donchin and Coles, 1988; Feuerriegel et al., 2018; Kolossa et al., 2012; Mars et al., 2008; Melloni et al., 2011; Meyniel, Maheu and Dehaene, 2016; Squires et al., 1976; Summerfield et al., 2011), whereas in the Bekinschtein et al.’s (2009) study, local vs. global effects refer to within- vs. between-chunks effects. The Bekinschtein et al.’s 2009 study says nothing about how many items or chunks are necessary to observe between-chunk effects.

Third, we previously discussed the link with Bekinschtein et al.’s 2009 study in two distinct paragraphs, which was not ideal. We have now assembled this point into only one paragraph as follows:

“A second finding from the “local-global” paradigm is that late brain responses (the P300) are sensitive to the rarity of patterns over the entire block. […] Testing the automaticity and task-dependence of our results will require further experiments.”

Last, thank you for pointing to us the paper by Garrido, Sahani and Dolan, 2012 which is highly relevant to our work. We now cite it (Garrido, Sahani and Dolan, 2013). Our results are indeed fully compatible with their findings as we indeed also find that early responses are modulated by a global statistic. However, we cannot speculate about the effect of the variance of the distribution in our case as, with Bernoulli processes (governing the generation of discrete events), the variance is confounded with the strength of the probabilistic bias (extreme probabilities induce lower variance).